# A trefoil knot self-templated through imination in water

Ye Lei[1,6], Zhaoyong Li[1,2,6], Guangcheng Wu[1], Lijie Zhang[3], Lu Tong[1], Tianyi Tong[4], Qiong Chen[1], Lingxiang Wang [1], Chenqi Ge[1], Yuxi Wei[1], Yuanjiang Pan[1], Andrew C.-H. Sue [4✉], Linjun Wang [1,2✉], Feihe Huang [1,5✉] & Hao Li [1,5✉]

The preparation of topologically nontrivial molecules is often assisted by covalent, supramolecular or coordinative templates that provide spatial pre-organization for all components. Herein, we report a trefoil knot that can be self-assembled efficiently in water without involving additional templates. The direct condensation of three equivalents of a tetraformyl precursor and six equivalents of a chiral diamine produces successfully a [3 + 6] trefoil knot whose intrinsic handedness is dictated by the stereochemical configuration of the diamine linkers. Contrary to the conventional wisdom that imine condensation is not amenable to use in water, the multivalent cooperativity between all the imine bonds within the framework makes this trefoil knot robust in the aqueous environment. Furthermore, the presence of water is proven to be essential for the trefoil knot formation. A topologically trivial macrocycle composed of two tetraformyl and four diamino building blocks is obtained when a similar reaction is performed in organic media, indicating that hydrophobic effect is a major driving force behind the scene.

[1] Department of Chemistry, Zhejiang University, Hangzhou 310027, PR China. [2] Key Laboratory of Excited-State Materials of Zhejiang Province, Zhejiang University, Hangzhou 310027, PR China. [3] Hangzhou Institute of Advanced Studies, Zhejiang Normal University, Hangzhou 311231, PR China. [4] College of Chemistry and Chemical Engineering, Xiamen University, Xiamen 361005, PR China. [5] ZJU-Hangzhou Global Scientific and Technological Innovation Center, Hangzhou 310027, PR China. [6] These authors contributed equally: Ye Lei, Zhaoyong Li. ✉email: Andrewsue@xmu.edu.cn; ljwang@zju.edu.cn; fhuang@zju.edu.cn; lihao2015@zju.edu.cn

Like their macroscopic counterparts, such as fishing nets and woven fabrics[1], molecular knots[2] are ubiquitous in the microscopic world. For example, flexible polymer chains[3,4] are known to form entanglements. Knots can also be observed in assorted biological systems such as DNA and proteins[5–7]. In some cases, these topologically complicated architectures are related to their biological activity[5]. In order to unravel the formation mechanism of these knots, as well as building a connection between functions and topology, synthesizing artificial knots in water, the medium for life, needs to be exploited. One commonly used strategy to construct such structures relies on the templation of transition metal cations, whose specific coordination geometries help to pre-organize the building blocks into the corresponding intermediates with crossing points. A variety of molecules with complex topology have been synthesized efficiently following this pathway, including trefoil knots[8–13], pentafoil knots[14–16], Borromean rings[17,18], Solomon links[19–21], Star of David catenanes[22–24] as well as eight-crossing molecular knots[25]. In comparison, the number of topologically nontrivial molecules synthesized without the involvement of transition metal templates is much smaller[26–32]. Feigel et al.[26] employed amino acid precursors in synthesizing a trefoil knot, whose formation is driven by hydrogen bonding between the amide building blocks. The Itami group[27,28] synthesized an all-benzene trefoil knot beautifully by using a removable silicon template to drive the formation of intertwined intermediates. Even though these methods represent milestones in the synthesis of topologically non-trivial molecules, their (further study) practical use is limited owing to the tedious multistep procedures and low yields. On the other hand, instead of employing external templates, the hydrophobic effect[31–34] can be used as the main driving force to fold the corresponding building blocks into the desired topologically nontrivial structures. When coupled with dynamic covalent bond formation, this approach enables the targeted trefoil knots to be obtained efficiently in one-pot reactions as the most thermodynamically favored products. This method has been demonstrated successfully by Sanders[34], who took advantage of the dynamic nature[35–39] of disulfide bond[40], as well as Cougnon[32], who employed hydrazone[41] formation reaction, for the synthesis of trefoil knots in water.

The dynamic covalent chemistry associated with imine formation[42–45], which involves aldehyde and amine precursors, is highly accessible synthetically and environmentally friendly. The labile nature of imine in the presence of water implies that it is not amenable to use in aqueous media. In the literature, it is reported that imine condensation was employed in the self-assembly of various micelles[46–48], dynamers[49], and supramolecular polymers[50,51]. In these cases, these polymeric structures act as the shelters for the imine bonds, protecting the latter from hydrolysis. The examples of using imine condensation in the self-assembly of small molecules with well-defined structures are much rarer. It has been shown[52–55] that a self-assembled molecule, whose building blocks are connected via multiple imine bonds, can be robust enough to stay stable even in pure water. By exploiting this so-called multivalency-enhanced stability of imine and hydrophobic effect altogether, we envision that the self-assembly of topologically complex molecules in water can be achieved.

Here, we show that a trefoil knot is obtained as the predominant product by condensing three equivalents of a tetraformyl precursor and six equivalents of a chiral bisamine in water. The formation of the trefoil knot is highly enantioselective. That is, the bisamines with *RR* and *SS* stereocenters exclusively produce the left-handed (*M*) and right-handed (*P*) trefoil knots, respectively. When a racemic mixture of the bisamine linker is used in the reaction, narcissistic self-sorting occurs, yielding a pair of enantiomeric trefoil knots, each of which contains six linker units with the same stereochemical configuration. When a similar reaction is performed in an organic media (Me₂SO or MeCN) instead of water, the resulting product changes to a topologically trivial macrocycle assembled from two equivalents of the tetraformyl precursor and four equivalents of the chiral bisamine. This result indicates unambiguously that hydrophobic effect is the major driving force in the formation of these trefoil knots.

## Results

**Self-assembly of the trefoil knot.** A dicationic tetraaldehyde precursor $1^{2+}\cdot2Br^-$ and $(1S,2S)$-(+)-1,2-diaminocyclohexane (($SS$)-CHDA) were combined (Fig. 1A) in a 1:2 ratio in $H_2O$ (1 mL). Here, $H_2O$ was employed instead of $D_2O$, in order to avoid deuteration. After the mixture was heated at 80 °C for 12 h, the solution was cooled down to room temperature. $CD_3SOCD_3$ (0.33 mL) was then added to the aqueous solution in order to perform $^1H$ NMR spectroscopic experiments. Both $^1H$ NMR spectrum (Fig. 2C) and ESI-HRMS (Supplementary Fig. S14) were recorded. ESI-HRMS reveals that a product, namely $S$-$2^{6+}$, containing three equivalents of $1^{2+}$ and six equivalents of ($SS$)-CHDA, was obtained as the major product. Two predominant peaks corresponding to $[2 - H]^{5+}$ and $[2 + Br]^{5+}$ were observed in the mass spectrum (Supplementary Fig. S14). The $^1H$ NMR spectrum shows a set of sharp resonances (Fig. 2C), indicating that the structure of $S$-$2^{6+}$ is highly symmetrical. The assignments were confirmed by two-dimensional NMR spectroscopy (Supplementary Figs. S6–S13). The resonance corresponding to the proton *f* in the biphenyl protons undergoes a significant upfield shift of around –2.4 ppm compared to the corresponding resonance in the precursor $1^{2+}$ (Fig. 2B). This result indicates that the biphenyl unit is buried deeply within the framework of $S$-$2^{6+}$ and therefore experiences a shielded magnetic environment. Only one resonance was observed corresponding to each proton in the biphenyl units including either *g* or *f*, indicating that the biphenyl units in the framework of $S$-$2^{6+}$ undergo fast circumvolution around the –biphenyl– axis on the $^1H$ NMR timescale. No other products in substantial amounts were observed in the $^1H$ NMR spectrum. The 2D DOSY spectrum (Fig. 2D) demonstrates that all resonances of $S$-$2^{6+}$ have the same diffusion coefficient. By using the Stokes–Einstein equation, the hydrodynamic diameter of $S$-$2^{6+}$ was calculated to be ~2.6 nm. In the framework of $S$-$2^{6+}$, the protons *e/e'* become diastereotopic in the $^1H$ NMR spectrum, revealing the chirality of the knot. In addition, the resonances corresponding to protons including either *c/c'* or *h/h'* also split into two peaks. This observation results from the fact that the two imine bonds grafted on a phenyl unit adopt two different orientations.

**Two-dimensional NMR spectroscopic characterization.** In order to shed more light on the structure of $S$-$2^{6+}$, its NOESY spectrum (Fig. 2E) was recorded. NOE signals were observed between the biphenyl protons (namely *g* and *f*), and the pyridinium or the phthalaldehyde protons including *c*, *c'*, *b*, and *h'*. The latter group of protons are believed to be relatively remote from the central biphenyl units. The appearance of NOE signal thus confirms that within the framework of $S$-$2^{6+}$, the building blocks form an intertwined architecture and have close contacts. NOE signals were also observed between the protons *c'* and *h'*, as well as *h* and *d*. These observations indicate that the two imine units in each phthalaldehyde residue have two different orientations relative to the central proton *d*—namely *endo* and *exo*. See the molecular formula in Fig. 2E. Given that the $^1H$ NMR spectrum of $S$-$2^{6+}$ exhibits only one set of resonances, we

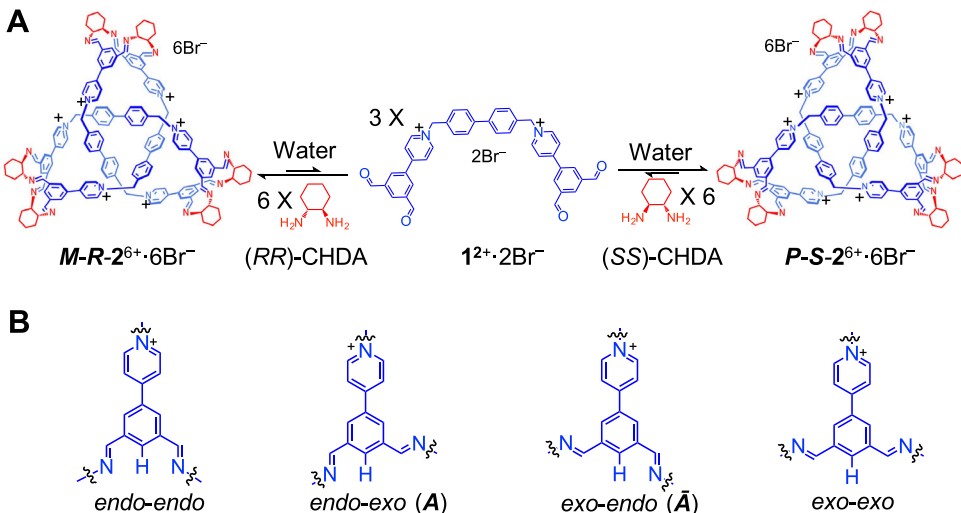

**Fig. 1 Structural formulae of a pair of enantiomeric trefoil knots $R$-$2^{6+}$·6Br$^-$, $S$-$2^{6+}$·6Br$^-$ and the corresponding precursors including $1^{2+}$·2Br$^-$ and CHDA. A** $R$-$2^{6+}$·6Br$^-$ and $S$-$2^{6+}$·6Br$^-$ were self-assembled as the predominant products, by condensing the tetraformyl precursor $1^{2+}$·2Br$^-$ and a chiral bisamine namely either ($RR$)-CHDA or ($SS$)-CHDA in aqueous media. DFT calculation indicated that the most thermodynamically favored products of $R$-$2^{6+}$ and $S$-$2^{6+}$ adopt intrinsic left-handed ($M$) and right-handed ($P$) configurations, respectively. **B** Each 1,3-bisiminophenyl unit in the $1^{2+}$ residues of the trefoil knot $M$-$R$-$2^{6+}$ and $P$-$S$-$2^{6+}$ could adopt four possible conformations including *endo-endo*, *endo-exo*, *exo-endo* and *exo-exo*, defined by the positions of each imino units relative to the central proton. NOESY spectrum (Fig. 2E) indicated that among these four conformations, only *endo-exo* and *exo-endo* were observed in the trefoil knots $R$-$2^{6+}$ and $S$-$2^{6+}$. Also, because the corresponding $^1$H NMR spectrum of either $M$-$R$-$2^{6+}$ or $P$-$S$-$2^{6+}$ only show a set of resonances, it is reasonable to propose that within the framework of each trefoil knot either $M$-$R$-$2^{6+}$ or $P$-$S$-$2^{6+}$, all of the six 1,3-bisiminophenyl units adopt the same conformation. DFT calculation indicated that the most thermodynamically favored conformations of $M$-$R$-$2^{6+}$ and $P$-$S$-$2^{6+}$ are (*endo-exo*)$_6$ and (*exo-endo*)$_6$, which are denoted as $A$ and $\bar{A}$, respectively.

propose that the six bisimino residues in the framework of $S$-$2^{6+}$ have the same conformation, either (*endo-exo*)$_6$ or (*exo-endo*)$_6$, which is denoted as $A$ or $\bar{A}$ (Fig. 1B), respectively.

**Self-assembly in organic media**. It is noteworthy that water is indispensable in the self-assembly of $S$-$2^{6+}$, indicating that hydrophobic effect is the major driving force of the knot formation. Condensing $1^{2+}$·2Br$^-$ and ($SS$)-CHDA in D$_2$O produces the same product—namely $S$-$2^{6+}$—as that occurs in the water-organic mixture namely H$_2$O/CD$_3$SOCD$_3$ (v/v, 3:1). The only difference is that, in D$_2$O, the relatively more acidic protons $a$ and $e/e$' undergo deuteration, which leads to disappearance of the corresponding resonances in the $^1$H NMR spectrum (Supplementary Fig. S24a). When the solvent was switched to purely organic media, the self-assembly yielded a different product. That is, when a 1:2 mixture of $1^{2+}$·2Br$^-$ (2 mM) and ($SS$)-CHDA (4 mM) were combined in CD$_3$SOCD$_3$, a macrocycle $S$-$3^{4+}$·4Br$^-$ was obtained as the major product containing two tetraformyl and four bisamino precursors, as indicated by the corresponding $^1$H NMR spectrum (Fig. 3) and ESI-HRMS (Supplementary Fig. S22). When $1^{2+}$·2PF$_6^-$ was used for the self-assembly in CD$_3$CN, a macrocycle $S$-$3^{4+}$·4PF$_6^-$ was obtained. In contrast with the trefoil knot $S$-$2^{6+}$ that was self-assembled as the only observable product in water, the formation of $S$-$3^{4+}$ was accompanied with a library of larger oligomeric or polymeric byproducts, which were observed as the minor products in the corresponding $^1$H NMR spectra recorded in both CD$_3$SOCD$_3$ and CD$_3$CN. Performing counterion exchange to $S$-$3^{4+}$·4PF$_6^-$ yielded $S$-$3^{4+}$·4Br$^-$ in a 47% yield. See details in the SI. Once dissolved in an aqueous phase, $^1$H NMR spectroscopic studies (Supplementary Fig. S31) indicated that the macrocycle $S$-$3^{4+}$ underwent transformation gradually into $S$-$2^{6+}$, driven by hydrophobic effect. At room temperature, it took a few months to finish the transformation, while at 80 °C, the structural change was nearly complete within no longer than 12 h. We recorded the NOESY

spectrum (Supplementary Fig. S21) of $S$-$3^{4+}$·4Br$^-$, which was in sharp contrast to that of $S$-$2^{6+}$·6Br$^-$. Firstly, no NOE signals were observed between the protons in the biphenyl units and those protons in either the pyridinium or phthalaldehyde residues. This observation is consistent with our hypothesis that, unlike the structure of $S$-$2^{6+}$, $S$-$3^{4+}$ does not have an intertwined architecture. Secondly, a NOE signal was observed between two imine protons $h$ and $h$', which was not observed in the case of the trefoil knot. This observation indicates that both of the two imine bonds in a phthalaldehyde residue of $S$-$3^{4+}$ orientate in an *exo-exo* manner (Fig. 1B), so that a proton $h$ is relatively close to a proton $h$' in an adjacent phthalaldehyde residue. See the red two-headed arrow in Fig. 3.

**Chirality of the trefoil knot**. The enantiomer of $S$-$2^{6+}$·6Br$^-$, namely $R$-$2^{6+}$·6Br$^-$, was also self-assembled successfully by condensing a 1:2 mixture of $1^{2+}$·2Br$^-$ and ($RR$)-CHDA, which is the enantiomer of ($SS$)-CHDA, in either D$_2$O or H$_2$O. When a racemic mixture of bisamino precursors was used in self-assembly, narcissistic self-sorting occurred. That is, a 1:1:1 mixture of $1^{2+}$·2Br$^-$, ($RR$)-CHDA, and ($SS$)-CHDA in an aqueous media produced a racemic mixture of $S$-$2^{6+}$·6Br$^-$ and $R$-$2^{6+}$·6Br$^-$, whose $^1$H NMR spectrum is identical to each of the individual chiral trefoil knots namely either $S$-$2^{6+}$·6Br$^-$ or $R$-$2^{6+}$·6Br$^-$. Such narcissistic self-sorting behavior was also observed by other scientists such as Sanders[34]. Replacing CHDA with a more flexible diamino linker, namely ethylenediamine, led to the failure of trefoil knot formation. This experiment indicated that restricted *gauche* conformation of the two amino units in CHDA is of importance in suppressing the entropy loss in knot formation. The importance of *gauche* effect in determining the distribution of self-assembly products was also reported by other scientists such as Cougnon[55]. The circular dichroism (CD) spectra of $S$-$2^{6+}$·6Br$^-$, $R$-$2^{6+}$·6Br$^-$, as well as the racemic mixture were recorded (Fig. 4). $S$-$2^{6+}$·6Br$^-$ (red trace) and $R$-$2^{6+}$·6Br$^-$ (black trace) have mirror-like CD signals, while the

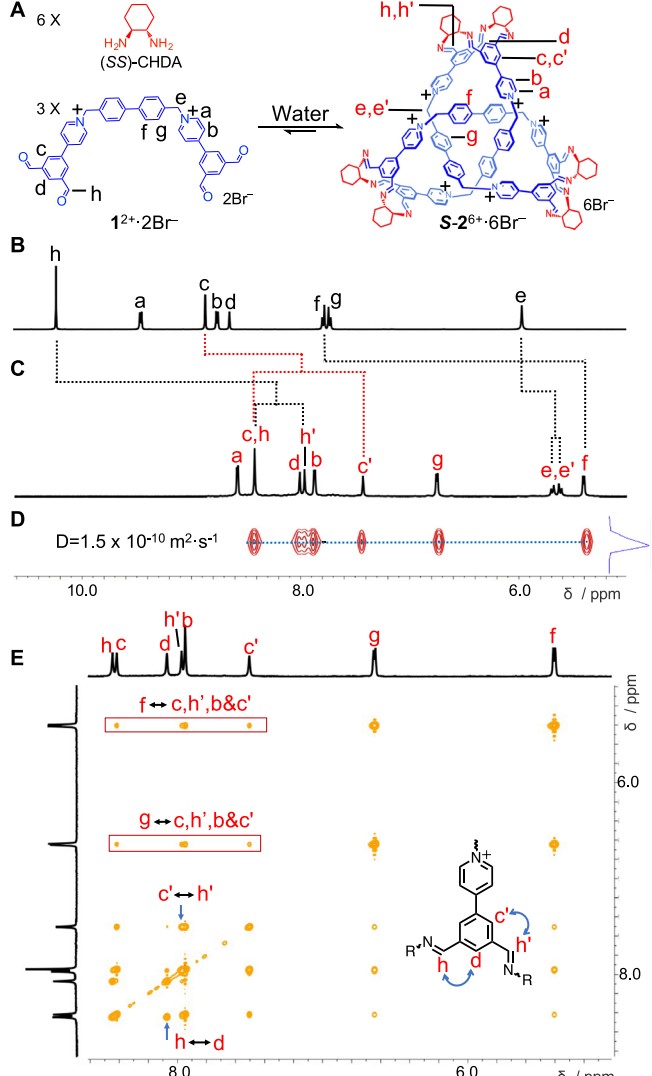

**Fig. 2 [1]H NMR spectroscopic characterization of S-2[6+]·6Br[−]. A** Structural formulae of **S-2[6+]·6Br[−]**. The partial [1]H NMR spectra (600 MHz, 298 K) of **B** the precursor **1[2+]·2Br[−]** recorded in CD₃SOCD₃ and **C S-2[6+]·6Br[−]** recorded in H₂O/CD₃SOCD₃ (v/v, 3:1). The sample used in **C** was obtained by heating a solution of **1[2+]·2Br[−]** (3 mM) and (SS)-CHDA (6 mM) in H₂O (1 mL) at 80 °C for 12 h, followed by adding CD₃SOCD₃ (0.33 mL) into the solution. **D** 2D DOSY spectrum (298 K, D₂O/CD₃SOCD₃ (v/v, 3:1)) of **S-2[6+]·6Br[−]**, which was prepared in the same procedure as that used in **C**, using D₂O instead of H₂O. The protons "e" and "a" underwent deuteration and the corresponding resonances disappeared. **E** The partial 2D NOESY spectrum (298 K, D₂O/CD₃SOCD₃ (v/v, 1:1)) of **S-2[6+]·6Br[−]**. NOE signals were clearly observed between the biphenyl protons (f and g) and those in either the pyridinium or the phthalaldehyde residues (including c, c', b and h'), which are encircled with red rectangles. NOE signals were also observed between the protons c' and h', as well as h and d, indicating the two imine units orientate in exo and endo manners respectively with respect to the central proton d.

racemic mixture (blue trace) of **S-2[6+]·6Br[−]** and **R-2[6+]·6Br[−]** is CD-silent. It is noteworthy that the approach of using a building block with stereo chiral centers to dictate the handedness of a topological complex molecule was also reported by other scientist[25,34], including Sauvage[56].

**Theoretical calculations.** Several trails to obtain diffraction grade single crystals of **S-2[6+]·6Br[−]** and **R-2[6+]·6Br[−]** as well as the

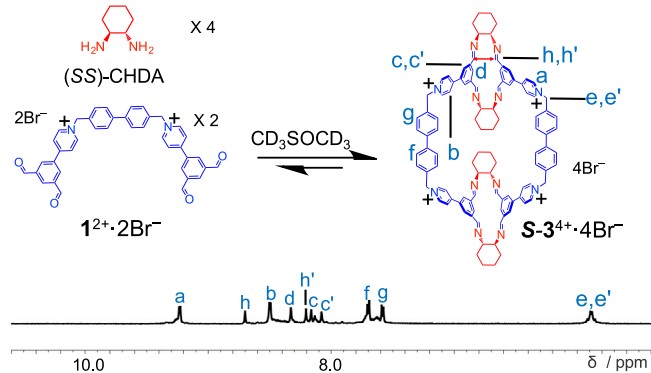

**Fig. 3 Partial [1]H NMR spectra (600 MHz, 298 K, CD₃SOCD₃) of S-3[4+]·4Br[−]. S-3[4+]·4Br[−]** was obtained by heating a 1:2 mixture of **1[2+]·2Br[−]** (2 mM) and (SS)-CHDA (4 mM) at 80 °C for 12 h in CD₃SOCD₃.

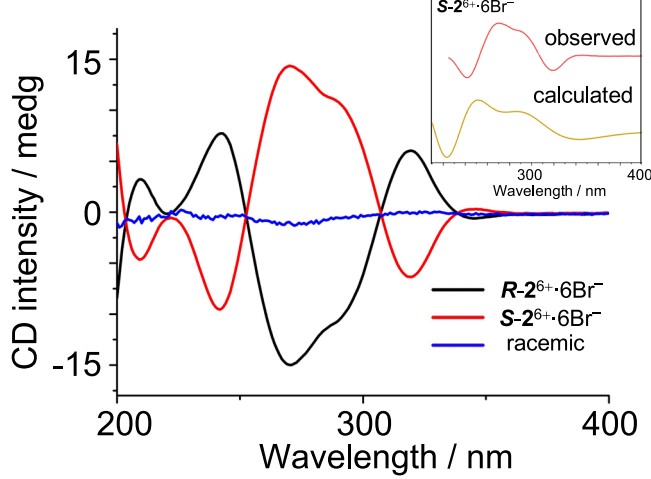

**Fig. 4 CD spectra of the product (0.04 mM) S-2[6+]·6Br[−] (red trace), R-2[6+]·6Br[−] (black trace), and their racemic mixture (blue trace) recorded in water.** The racemic cage mixture was self-assembled by condensing a racemic CHDA and the tetraaldehyde **1[2+]·2Br[−]** in water. Insert: the CD spectra of **S-2[6+]·6Br[−]** based on experiment (red trace), and theoretical calculated structure (yellow trace) which was obtained based on the optimized structure of the knot **P-Ā-S-2[6+]** shown in Fig. 5. **P-Ā-S-2[6+]** was considered as the most thermodynamically favored isomer among the four possible conformational diastereomers.

macrocycle **S-3[4+]·4PF₆[−]** have so far proved unsuccessful. Hence, density functional theory (DFT) calculation (see details in the SI) was used to shed more light on the architectures of these trefoil knots. Theoretically, the trefoil knot **2[6+]** might have a number of diastereoisomers due to three asymmetrical factors. Firstly, the CHDA linkers could be either (SS) or (RR), depending on the bisamino precursor used in self-assembly. Secondly, the framework of a trefoil knot could be intrinsically right-handed (P) or left-handed (M), depending on the intertwining direction of the building blocks. Thirdly, as mentioned before, NOE signals indicated that each of the six phthalaldehyde residues in the trefoil knot should have the same conformation, namely either (endo-exo)₆ or (exo-endo)₆, which is denoted as **A** or **Ā** (Fig. 1B), respectively. The implication is that, the trefoil **S-2[6+]** is supposed to have four diastereoisomers, including **P-Ā-S-2[6+]**, **M-Ā-S-2[6+]**, **M-A-S-2[6+]**, and **P-A-S-2[6+]**, whose enantiomers are **M-A-R-2[6+]**, **P-A-R-2[6+]**, **P-Ā-R-2[6+]**, and **M-Ā-R-2[6+]**, respectively. DFT calculations (Fig. 5) indicated that the relative free energy is 0, 17.6, 25.6, and 39.4 kcal/mol in the case of **P-Ā-S-2[6+]/M-A-R-2[6+]**, **M-**

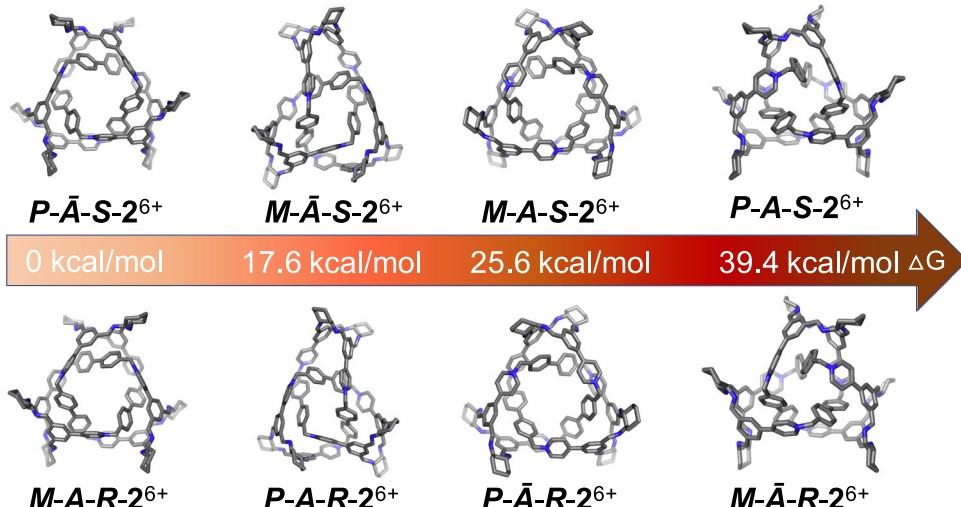

**Fig. 5 Optimized structures of different possible conformational diastereomers of $S$-$2^{6+}$ (top) and $R$-$2^{6+}$ (bottom) with the relative free energies.** The calculation was performed by using DFT calculations at the level of BP86-D3 functional and 6-311G(d) basis set. The result indicates that $P$-$\bar{A}$-$S$-$2^{6+}$ and its enantiomer namely $M$-$A$-$S$-$2^{6+}$ are the most favored products among the diastereomers of $R$-$2^{6+}$ and $S$-$2^{6+}$.

$\bar{A}$-$S$-$2^{6+}$/$P$-$A$-$R$-$2^{6+}$, $M$-$A$-$S$-$2^{6+}$/$P$-$\bar{A}$-$R$-$2^{6+}$, and $P$-$A$-$S$-$2^{6+}$/$M$-$\bar{A}$-$R$-$2^{6+}$, respectively, given that enantiomers should be thermodynamically equal in achiral solvent. The implication is that $P$-$\bar{A}$-$S$-$2^{6+}$/$M$-$A$-$R$-$2^{6+}$ with the lowest stabilization energy are the most favored products among the four diastereoisomers. Based on theoretically optimized structures of all four diastereoisomers namely $P$-$\bar{A}$-$S$-$2^{6+}$, $P$-$\bar{A}$-$R$-$2^{6+}$, $P$-$A$-$R$-$2^{6+}$, and $P$-$A$-$S$-$2^{6+}$, we were able to calculate their electronic CD spectra (Supplementary Fig. S33d). However, only the spectrum of $P$-$\bar{A}$-$S$-$2^{6+}$ (Fig. 4, insert, yellow trace) matched well with the experimental one (Fig. 4, red trace), supporting our hypothesis that among the four isomers, $P$-$\bar{A}$-$S$-$2^{6+}$ is self-assembled selectively as the most thermodynamically favored product. The diameter of the optimized structure of $P$-$\bar{A}$-$S$-$2^{6+}$ is measured to be about 26.4 Å, which is consistent with the experimental value (i.e., 26 Å) from DOSY experiment.

## Discussion
By taking advantage of the marriage of hydrophobic driving force and the dynamic behavior of imine formation, a pair of enantiomers of purely covalent chiral trefoil knots was self-assembled as the predominant product in a one-pot manner in aqueous media. Thanks to multivalency, the trefoil knot containing 12 imine bonds is robust enough in both water and water-organic mixture (a 3:1 mixture of water and $CD_3SOCD_3$). In organic media including either $CD_3SOCD_3$ or $CD_3CN$, self-assembly went to another pathway—namely that a topologically trivial macrocyclic compound was self-assembled. This result indicated unambiguously that hydrophobic effect is the major driving force of the formation of trefoil knot. The intrinsic handedness of the trefoil knot is dictated precisely by the stereo chirality of the bisamino precursors, namely that the biamines with $SS$ or $RR$ stereo-centers led to exclusively the formation of trefoil knots with $P$ or $M$ handedness, respectively. The success in self-assembly of trefoil knot with controllable handedness improved our fundamental understanding on how nature uses small chiral building blocks to create topologically complex architectures with amplified chirality. This work further supports that imine-based dynamic covalent chemistry can be performed in the presence of water, which would encourage chemists to self-assemble another topologically complex molecule in aqueous media. The water compatibility of this trefoil knot also encourages chemists to explore its potential functions in biological media.

## Methods
**Self-assembly of $S$-$2^{6+}$·6Br⁻.** A 1:2 mixture of $1^{2+}$·2Br⁻ (1.5 mg, 0.002 mmol) and (1$S$,2$S$)-(+)-1,2-diaminocyclohexane (0.46 mg, 0.004 mmol) was dissolved in $D_2O$ (0.6 mL). The corresponding reaction mixture was heated at 80 °C for 12 h. $S$-$2^{6+}$ was self-assembled as the only observable product in the corresponding ¹H NMR spectrum, without further manipulation. The attempts to isolate the trefoil knot $S$-$2^{6+}$ from its aqueous solution via counterion exchange proved unsuccessful, probably because imine bonds underwent degradation during these processes. We also tried to fix the architecture of the trefoil knot by means of imine reduction. Such trials were not successful either, because under such conditions, the cationic pyridinium building blocks became labile.

**Synthesis of $S$-$3^{4+}$·4Br⁻.** A 1:2 mixture of $1^{2+}$·2PF₆⁻ (30 mg, 0.034 mmol) and (1$S$,2$S$)-(+)-1,2-diaminocyclohexane (7.8 mg, 0.068 mmol) was dissolved in $CH_3CN$ (18 mL). The corresponding reaction mixture was heated at 70 °C for 12 h. After cooling down the reaction solution to room temperature, tetrabutylammonium bromide (TBA⁺·Br⁻) (100 mg) was then added to the solution, after which a white solid was collected by filtration and washed with $CH_2Cl_2$ three times, yielding $S$-$3^{4+}$·4Br⁻ (30 mg, 47%) as a white solid. It is noteworthy that raising the concentrations of the precursors namely $1^{2+}$·2PF₆⁻ and (1$S$,2$S$)-(+)-1,2-diaminocyclohexane would lead to the formation of oligomeric and polymeric byproducts.

**Structural characterization of $S$-$2^{6+}$·6Br⁻ and $S$-$3^{4+}$·4Br⁻.** Nuclear magnetic resonance (NMR) spectra were recorded at ambient temperature using Agilent DD2 600 spectrometers, with working frequencies of 600 and 150 for ¹H and ¹³C respectively. Chemical shifts are reported in ppm relative to the residual internal non-deuterated solvent signals (for proton NMR, $D_2O$: $\delta = 4.70$ ppm, $CD_3SOCD_3$: $\delta = 2.50$ ppm). High-resolution mass spectra were recorded on a Fourier transform ion cyclotron resonance mass spectrometry. CD spectra were recorded on a Circular Dichroism Spectrometer (Chirascan V100, Applied Photophysics Ltd).

**Theoretical calculations**. The trefoil knots were optimized by using the DFT at the BP86-D3/6-311G(d) level with the Gaussian 16 package[57]. The solvent effect of water was included with the polarizable continuum model using solvent-accessible surface. The UV–Vis and electronic CD spectra for the optimized geometries were predicted by the time-dependent DFT at the wB97XD/6-311G(d) level with 500 states by the Gaussian 16 package[57]. The solvent effect was also considered. Both the UV–Vis and CD spectra were broadened using Gaussian functions by Multiwfn[58]. The full width at half maximum was set as 0.8 eV.

## Data availability
The authors declare that all other data supporting the findings of this study are available from the article and its Supplementary Information. Cartesian coordinates are provided in Supplementary Data file 1. Additional data are available from the authors upon request.

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

## Acknowledgements

This work was supported by the National Natural Science Foundation of China (No. 21922108 to H.L.; No. 21922305 to L.W.; No. 21927810 to Y.P.). H.L. and F.H. also want to thank the support from the Starry Night Science Fund of Zhejiang University Shanghai Institute for Advanced Study (SN-ZJU-SIAS-006). L.W. also acknowledges support from the High-Performance Computing Center in the Department of Chemistry, Zhejiang University.

## Author contributions

H.L. conceived the project. Y.L., Z.L., G.W., L.Z., L.T., Q.C., L.W. (Lingxiang Wang), A.C.-H.S., L.W. (Linjun Wang), F.H. and H.L. prepared the manuscript. Y.L., G.W., L.T., Q.C., C.G., and Y.W. synthesized the molecules studied in this work. Y.L, G.W., and Y.P. performed characterization of the key compounds. Y.L. built the primary structures. T.T. made some calculations. Under the supervision of L.W. (Linjun Wang), Z.L. and L.Z. performed the major theoretical calculations and analysis. All authors discussed the results and commented on the manuscript.

## Competing interests

The authors declare no competing interests.
