## [Peer Review File · Nature Communications]

Reviewer #1 (Remarks to the Author):

In this interesting article, Li and co-workers describe the discovery that dynamic imine chemistry can be used to self-assemble a molecular trefoil knot in water. A dicationic tetraaldehyde **1** and a trans-1,2-diaminocyclohexane (with point chirality of either R,R or S,S) mixed in a 1:2 ratio self-assembled into the trefoil knot product after heating overnight, and the topologically complex product was then directly analysed by CD, 1D and 2D NMR spectroscopy and MS. Very interestingly, the hydrophobic effect was crucial for the success of the reaction, as performing the same imine condensation in organic solvent led to the formation of a topologically trivial macrocycle instead. The knot also formed with single handedness (as dictated by the point chirality in the diaminocyclohexane building block) and underwent narcissistic self-sorting into a 1:1 mix of left- and right-handed knots upon use of racemic diaminocyclohexane. As the authors state, it is very difficult to obtain molecular knots in aqueous solution, and even more so from moisture-sensitive imine bonds. Thus, this is clearly an impressive and important piece of work, and the use of the hydrophobic effect as a tool to direct self-assembly of knots provides a compelling alternative to the traditional metal template approach. The efficiency of the assembly is seemingly also quite good (58% yield is a number reported in the SI, but more on this later), and there are a number of very interesting and impressive features in this system that makes it a highly compelling piece of supramolecular chemistry.

With that said, the main drawback of this system (like with any of the more serendipitous self-assembly methods to make knots and links) is that the system seems to lack some flexibility, and a very specific set of building blocks need to be used to achieve the intended synthesis. This will naturally limit the more general conclusions that can be drawn from a broader topological perspective from this class of molecules. The system looks a bit like the imine sibling of Sanders's famous disulfide trefoil knot (ref 34) which also assembles efficiently in water using dynamic covalent chemistry and the hydrophobic effect. Sanders' system is very elegant (just as like this piece of work), but has not seen any use anywhere outside of the original report. The authors here state at the end that they will attempt to use this new imine molecular knot for asymmetric catalysis. While this would be great, I highly doubt that would be successful given the apparent issues with getting the knot pure/isolated, and I don't really expect the molecule to find much practical use in the same way that Trabolsi has used his metal templated knot system (see ACS Catal., 2019, 9, 1907; Chem. Sci., 2019, 10, 5884) or Leigh has used his Ln-based knots (J. Am. Chem. Soc., 2016, 138, 13159; Angew. Chem. Int. Ed., 2018, 57, 10484; Nat. Chem., 2020, 12, 939).

Still, this is a beautiful structure, a competently performed study and there is more than enough to learn here in terms of design rules for knot synthesis and imine assembly in water that the paper should be publishable in Nature Communications. Some major scientific issues must however be addressed before that stage, but if the authors can address the following comments and concerns I would be delighted to recommend this paper for publication.

Big concerns:

- The paper does not do a particularly good job of explaining how the knot is prepared, nor in which yields. One has to go into the supporting information to see that the knot is actually never isolated, and that yields are only about 58%. This is not really acceptable reporting practice - it is never actually explained that the knot is not really isolated and 42% of the material does not form the intended assembly (so the main system is really only 58% pure, so to say). The authors should first of all make this much more apparent from the main text. Further, I think that at the very least the authors should try to reduce the imine bonds in the knot with NaBH₄, isolate the reduced compound in pure form and characterize it fully to get a proper yield. That could potentially also give some more spectral information on structure/topology that would be interesting. The authors also need discuss what else is forming in the system and why, as the focus of the paper

is phenomenological and not going into that sort of detail is quite an oversight considering the mechanistic/structural view that the paper takes.

- Definite structural proof in the form of XRD data is also missing for this knot, but the authors have instead employed a wide range of advanced spectroscopic techniques along with clever control experiments to demonstrate that the structure is what they propose. This methodology seems to be quite common nowadays, as a range of recent topologically complex molecules have been reported without crystal data (refs 31, 32 and 34 along with Nature, 2020, 584, 562). The evidence here is very compelling, but it is also conceivable (but unlikely) that the product could be a 2+1 [2]catenane in fast exchange (like the Hunter/Vögtle knot story - Angew. Chem. Int. Ed., 2000, 39, 1616). Did the authors check the MS/MS fragmentation pattern to confirm this is really a knot rather than a catenane? If not, that experiment could be good to perform. It's very unlikely that the structure is anything except a trefoil knot, but it is a simple experiment to rule out another explanation.

- The CD data of the knot enantiomers is nice, but a missing piece of data that would be very good to have for comparison is the CD of the enantiomers of the macrocycle 3. Have the authors measured this?

- Have the authors tried cooling the solution to freeze out the rotation of protons g and f? It would be very interesting to see the energetic barrier to this rotation, as it could tell a lot about the knot tightness.

- The references need some additional work. Nitschke's recent imine knot (Chem, 2021, 1534) should definitely be cited, as should some of the pioneering knot work from Vögtle (rather than the example from Feigel which is very interesting but was derived from Vögtle's lab). Also, an intro sentence say "In some cases, these topologically complicated architectures are related to their biological activity^{5,8-10}" but in ref 8 the topology has nothing to do with the biological activity and ref 9 and 10 don't concern biology at all (they're highly relevant papers to cite though, just not in this context).

Small comments:

- The term "Solomon knot" should be changed to "Solomon link".

- Sentence "By using the Stokes-Einstein equation, the hydrodynamic diameter of S-26+ was calculated to be approximately 2.6 nm" – maybe compare this value with the diameter calculated from the molecular model?

- Sentence "In the framework of S-26+, the protons e/e', c/c' and h/h' become diastereotopic in the 1H NMR spectrum, revealing the chirality of the knot." As per the authors own discussion later, isn't the h/h' splitting due to endo/exo effects rather than diastereotopicity?

- Sentence "This observation indicates unambiguously that S-34+ is a topologically trivial molecule." I don't agree – NOESY is notoriously sensitive to dynamics so a topologically complex molecule in fast exchange could still lack NOESY signals.

Rephrase?

- SI: DMF is not a very good internal standard due to its low boiling point (relatively speaking). Over longer time frames and at elevated temperatures you can lose a bit of DMF due to evaporation, hence giving you inaccurate readings.

Reviewer #2 (Remarks to the Author):

The manuscript submitted by Hao Li et al. reports the hydrophobically-driven synthesis of a trefoil knot. Overall, the study is well done and the conclusions appropriately drawn. My main issue with this manuscript concerns the novelty of the work. In my opinion, the most important findings of the study have already been reported elsewhere. If I go through the abstract:

(1) "Herein, we report a rare case of a trefoil knot that can be self-assembled efficiently

in water without involving additional templates." The hydrophobically-driven synthesis of trefoil knots and other topologically complex molecules have been reported in ref. 31, 32, 33 and 34.

(2) "The direct condensation of three equivalents of a tetraformyl precursor and six equivalents of a chiral diamine produced successfully a [3+6] trefoil knot whose intrinsic handedness is dictated by the stereochemical configuration of the diamine linkers." This effect is well-established. It has been known since the pioneering work of Sauvage (ACIE 2004, 43, 4482) and has been observed numerous times. Particularly relevant examples include Chem 2021, 7, 6, 1541 and ref. 13-14.

(3) "When a racemic mixture of the bisamine linker was used in the reaction, narcissistic self-sorting occurred, yielding a pair of enantiomeric trefoil knots, each of which contains six linker units with the same stereochemical configuration." The most relevant, previously published, example of this behaviour is perhaps ref. 34.

(4) "Contrary to the conventional wisdom that water is not compatible with the dynamic covalent chemistry of imine condensation, the multivalent cooperativity between all the imine bonds within the framework makes this trefoil knot robust in the aqueous environment." This argument was already the selling point of another article published last year by the same group (ref. 46). The idea can be traced back to the work of Warmuth on water soluble imine-based cryptophanes (ref 47-48). Along the same lines, the conclusion reads: "This work also overturns the commonly accepted precept that imine-based dynamic covalent chemistry cannot be performed in the presence of water, which would encourage chemists to self-assemble other topologically complex molecules in aqueous media." However, imine bonds were already used for the synthesis of interlocked molecules in pure water in Chem. Sci. 2018, 9, 1317, and the argument of this previous study was identical. This study also provides an analysis of the effect of gauche interactions on the assembly of interlocked molecules that seems relevant to the current findings (see "... indicated that restricted gauche conformations..." p. 6).

(5) "Furthermore, the presence of water was proven to be essential for the trefoil knot formation." See most of the references mentioned above.

What exactly do we learn from this paper that was not previously published? This manuscript presents an extremely interesting finding, but I believe the authors did not make the most of it and have confined their work to familiar ground. Therefore I would not recommend publication of this work, as it is now, at the level of Nature Communications.

This study has also raised a few questions:

(1) What is the effect of temperature on the process of self-assembly and on the hydrolysis of the imine bond (which is usually rapid at high temperature)? The data presented is rather scarce (Figure S22-24) and is not quantitative. It is difficult to draw any conclusion relative to the kinetics of formation of S-2 or to the kinetics of formation/hydrolysis of the imine bond.

(2) The variable temperature of the trefoil knot was only run up to 60°C while that of the macrocycle was run up to 110°C (Figure 23-24). Any reason for that?

(3) What is the effect of concentration of the assembly? The trefoil knot is more likely to hydrolyse at low concentration. Yet, the authors managed to measure a CD spectrum. I could not find the concentration at which this spectrum has been recorded, but I can imagine it's two to three orders of magnitude lower than that of the NMR experiments. Is the trefoil knot also stable in dilute conditions over time?

(4) Knowing the expertise of the Li group in anion recognition, it is somewhat surprising to see that the role of the counter ion in the assembly of S-2 has not been investigated. Based on the DFT calculations, it looks like the trefoil knot could possess a central cavity with several converging C-H hydrogen bond donors. I am not sure whether this cavity is large enough to accommodate bromide. Nevertheless, the observation that the trefoil knot flies with one and two bromides in MS (Figure S13), while the trivial macrocycle does not (Figure 21), suggests that two equivalents of bromides could be bound in the cavity (see Chem. Sci. 2016, 7, 2524 for a previous example). Does bromide (or other anions) bind to the knot? Does it template its formation?

(5) Imine bond hydrolysis strongly depends on pH. In the manuscript, there is no

mention of the pH of the solution during the reaction. Does pH affect the assembly?

(6) While I am convinced by the knotted topology of S-2, I wonder whether it is possible to be sure that S-2 is not a trivial macrocycle folded in a chiral conformation.

There are also a few mistakes that must be corrected before publication.

(1) p. 4: "the protons e/e', c/c' and h/h' become diastereotopic in the ^1H NMR spectrum, revealing the chirality of the knot". Protons e/e' are diastereotopic. Protons c/c' and h/h' are not diastereotopic, they are simply inequivalent because the two imine linkages adopt different conformations (exo and endo). This inequivalence has nothing to do with the chirality of the trefoil knot.

(2) p. 4 "Only one resonance was observed corresponding to proton g and f, indicating that the biphenyl units in the framework of S-26+ undergo fast circumvolution around the -biphenyl- axis on the ^1H NMR timescale." Whether it occurs rapidly or not, circumrotation around the biphenyl axis does not exchange the protons g and f and cannot explain this (incidental?) equivalence.

Reviewer #3 (Remarks to the Author):

From the perspective of theoretical calculation, this is a nice work to assist the experiment to explain the mechanism behind the experimental phenomenon. However, we know that for such a system, since the rotation of the dihedral Angle can produce many possible conformations, how can the author ensure that the present structure is representative of the real experimental system? Details are needed here, including how many possible structures are built and how to eliminate some of the impossible ones. In addition, the relative free energy is more meaningful than the relative energy, and the calculation of frequency can verify whether the structure is the local minimum point of the potential energy surface.

Letter of Response to the Reviewers' Comments

Reviewer #1 (Remarks to the Author):

Comment: In this interesting article, Li and co-workers describe the discovery that dynamic imine chemistry can be used to self-assemble a molecular trefoil knot in water. A dicationic tetraaldehyde **1** and a trans-1,2-diaminocyclohexane (with point chirality of either R,R or S,S) mixed in a 1:2 ratio self-assembled into the trefoil knot product after heating overnight, and the topologically complex product was then directly analysed by CD, 1D and 2D NMR spectroscopy and MS. Very interestingly, the hydrophobic effect was crucial for the success of the reaction, as performing the same imine condensation in organic solvent led to the formation of a topologically trivial macrocycle instead. The knot also formed with single handedness (as dictated by the point chirality in the diaminocyclohexane building block) and underwent narcissistic self-sorting into a 1:1 mix of left- and right-handed knots upon use of racemic diaminocyclohexane.

As the authors state, it is very difficult to obtain molecular knots in aqueous solution, and even more so from moisture-sensitive imine bonds. Thus, this is clearly an impressive and important piece of work, and the use of the hydrophobic effect as a tool to direct self-assembly of knots provides a compelling alternative to the traditional metal template approach. The efficiency of the assembly is seemingly also quite good (58% yield is a number reported in the SI, but more on this later), and there are a number of very interesting and impressive features in this system that makes it a highly compelling piece of supramolecular chemistry.

Response: We thank the reviewer for reading the paper so carefully and providing such a relatively positive comment.

Comment: With that said, the main drawback of this system (like with any of the more serendipitous self-assembly methods to make knots and links) is that the system seems to lack some flexibility, and a very specific set of building blocks need to be used to achieve the intended synthesis. This will naturally limit the more general conclusions that can be drawn from a broader topological perspective from this class of molecules. The system looks a bit like the imine sibling of Sanders' s famous disulfide trefoil knot (ref 34) which also assembles efficiently in water using dynamic covalent chemistry and the hydrophobic effect. Sanders' system is very elegant (just as like this piece of work), but has not seen any use anywhere outside of the original report. The authors here state at the end that they will attempt to use this new imine molecular knot for asymmetric catalysis. While this would be great, I highly doubt that would be successful given the apparent issues with getting the knot pure/isolated, and I don' t really expect the molecule to find much practical use in the same way that Trabolsi has used his metal templated knot system (see ACS Catal., 2019, 9, 1907; Chem. Sci., 2019, 10, 5884) or Leigh has used his Ln-based knots (J. Am. Chem. Soc., 2016, 138, 13159; Angew. Chem. Int. Ed., 2018, 57, 10484; Nat. Chem., 2020, 12, 939).

Response: The reviewer is fully correct in the statement that the synthesis of our trefoil knot obtained inspiration from the paper of Sanders, which was cited (ref 34). We also agree with the reviewer that,

success of both of these two systems needs the specific set of building blocks, which were sophisticatedly designed. However, it is noteworthy that even though our results did not provide a general strategy for synthesizing trefoil knots, it opens up opportunities to use metal-free systems to self-assemble topologically complex molecules. In addition, the success of our system also encourages chemists to use the marriage of imine and hydrophobic effect to prepare more complex molecules in water, a solvent that is considered as the “forbidden zone” of imine.

In terms of taking advantage of the handedness of trefoil knot for asymmetric catalysis, the reviewer is correct in claiming that it is very unlikely to use the current system to reach this purpose, because the trefoil knot is not isolated as a pure compound. This is because imine bond is rather labile in water. The truth is that, however, in the future we might use some more robust counterparts, such as hydrazone or oxime, for self-assembly, instead of imine. The future-developed systems could thus be very stable and can be isolated as chemically inert compounds. This is indeed the case in many of our systems containing hydrazone or oxime, which are very inert during counterion exchange or chromatographic purification. In addition, we could increase the sizes of building blocks, so that the future-obtained chiral knots would have larger sizes for guest accommodation.

Anyway, given that this future plan is still a blueprint and has a lot of uncertainties, we deleted the last sentence of the main-text, which used to claim that the trefoil knot might be used for chiral catalysis.

Comment: Still, this is a beautiful structure, a competently performed study and there is more than enough to learn here in terms of design rules for knot synthesis and imine assembly in water that the paper should be publishable in Nature Communications. Some major scientific issues must however be addressed before that stage, but if the authors can address the following comments and concerns I would be delighted to recommend this paper for publication.

Response: We thank the reviewer for the relatively positive comment. We are trying our best to address all the issues raised by the reviewer.

Comment: The paper does not do a particularly good job of explaining how the knot is prepared, nor in which yields. One has to go into the supporting information to see that the knot is actually never isolated, and that yields are only about 58%. This is not really acceptable reporting practice - it is never actually explained that the knot is not really isolated and 42% of the material does not form the intended assembly (so the main system is really only 58% pure, so to say). The authors should first of all make this much more apparent from the main text.

Response: In fact, synthesizing the trefoil knot is just like preparing a ^1H NMR spectrum sample, namely mixing the precursors in an NMR tube and heating the solution. Also, the trefoil knot is too labile to be isolated as solid-state sample, i.e., we did try to isolate its PF_6 salt by adding NH_4PF_6 in its aqueous solution, during which the trefoil knot underwent decomposition. To sum up, there are no special “tricks” in either preparation or purification. All we did was just “mixing”, “heating” and “waiting”.

In terms of the yields (see figure S26), it is noteworthy that the 58% is just the yield of the trefoil knot self-assembled in an aqueous/organic mixture containing 1:4 DMSO/ D_2O , by using an internal standard to calculate the relative concentrations of both the aldehyde precursor and the trefoil knot product. ^1H NMR spectra clearly indicated (Figure S26) that the yield of the trefoil knot decreases with the increase of the content of organic solvent. It is therefore reasonably hypothesized that in pure water (i.e., 0:10 DMSO/ D_2O), the yield of trefoil knot is higher than 58%. However, the attempts to quantitatively determine the yield of trefoil knot in pure water were unsuccessful. This is because the aldehyde precursor namely

$1^{2+} \cdot 2\text{Br}^-$ was just partially soluble in pure water. As a consequence, we cannot use an internal standard in the corresponding ^1H NMR spectra to determine the concentration of the aldehyde precursor, as we did in the organic/aqueous mixture where the aldehyde precursor was fully soluble. Another trial is to use high-dilution condition, e.g., 0.1 mM, to fully dissolve the aldehyde precursor. However, such trial was also unsuccessful, because in such a low concentration, imine formation reaction is not complete. Or expressed in another way, in pure water, the yield of the trefoil knot should be a value that is higher than 58%. In order to obtain the yield of $S\text{-}2^{6+}$ in pure water, we also found a precursor with a better solubility, namely $1^{2+} \cdot 2\text{CH}_3\text{COO}^-$. After the amine precursor namely (SS)-CHDA was added into the water solution of $1^{2+} \cdot 2\text{CH}_3\text{COO}^-$, we heated the solution at 80 °C. The yield was calculated to be 53% by using an internal standard, namely DMF. However, we discovered that $S\text{-}2^{6+} \cdot 2\text{CH}_3\text{COO}^-$ would undergo decomposition partially even at room temperature due to the basicity of CH_3COO^- , which means that the yields would be more than what we calculated.

We modified the main-text accordingly, to make these issues clearer.

Comment: Further, I think that at the very least the authors should try to reduce the imine bonds in the knot with NaBH_4 , isolate the reduced compound in pure form and characterize it fully to get a proper yield. That could potentially also give some more spectral information on structure/topology that would be interesting.

Response: We thank the reviewer for providing this suggestion. In fact, reducing imine is an often-used method to trap the self-assembled structures. However, our system contains six pyridinium units. When we used NaBH_4 to reduce the knot, it seemed that the pyridinium functional groups in the trefoil knot were reduced before the imine bonds. After pyridinium was reduced, the adducts lose its cationic nature and became insoluble in water. After reduction, we recorded the ^1H NMR spectrum of the aqueous solution (see the spectrum below), and we did not observe any resonances corresponding to the reduced trefoil knot. We also collected the precipitates and dissolved it in DMSO-d_6 . ^1H NMR spectrum of this precipitate also indicated that this solid is also a library of intractable mixture, instead of reduced trefoil knot.

Comment: The authors also need discuss what else is forming in the system and why, as the focus of the paper is phenomenological and not going into that sort of detail is quite an oversight considering the

mechanistic/structural view that the paper takes.

Response: First, as we mentioned before, in pure water, the self-assembly yield of the trefoil knot $S-2^{6+}$ to Br^- is much higher than 58%, implying that the yield of byproducts is lower than 42%. In fact, these byproducts were not even observable in the corresponding 1H NMR spectrum and mass spectrum. Or expressed in another way, the trefoil knot is the only product with an observable yield in pure water. We did not really observe anything else with substantial amount that can be characterized experimentally.

During the self-assembly of the trefoil knot before equilibrium, however, we did observe a [4+8] byproduct, namely a byproduct that is composed of four equivalents of the tetraaldehyde and eight equivalents of the bisamine. Mass spectrum clearly indicated the existence of this [4+8] product (see the spectrum above). The DOSY spectrum clearly indicated that the [4+8] byproduct has a slightly smaller diffusion coefficient compared to the trefoil knot (a [3+6] product). This DOSY result is thus perfectly consistent with the mass spectrum that, the byproduct is a [4+8] compound has a larger molecular weight compared to the trefoil knot.

In the 1H NMR spectrum, some of the resonances in this [4+8] byproduct underwent remarkable upfield shifts, in a similar manner as that occurred in the trefoil knot. We thus hypothesize that this [4+8] byproduct is either a Solomon link or a [2]catenane composed of two [2+4] macrocycles (see the structure below). However, it is noteworthy that, if the [4+8] product is a [2]catenane, the intramolecular circumvolution motion of each ring around the other should be relatively slow, due to the two bulky "speed bumps" (see the cyclohexanes in the structure). As a consequence, in the case of the [2]catenane, the resonances of the unit "inside" would differ from those of the unit "outside" (see the structure below). This hypothesis is inconsistent with the highly symmetric 1H NMR spectrum of the [4+8] product. To sum up, the [4+8] product should be a Solomon link.

It is noteworthy that the [4+8] byproduct is not a thermodynamically favored product. It will gradually transform into the trefoil knot within a few hours. Therefore, we were not able to isolate this [4+8] compound. And therefore, it cannot be characterized in a more conclusive manner or in its pure form. As a consequence, we chose to discuss its information here in the response letter and SI, instead of in the maintext.

Comment: Definite structural proof in the form of XRD data is also missing for this knot, but the authors have instead employed a wide range of advanced spectroscopic techniques along with clever control experiments to demonstrate that the structure is what they propose. This methodology seems to be quite common nowadays, as a range of recent topologically complex molecules have been reported without crystal data (refs 31, 32 and 34 along with Nature, 2020, 584, 562).

Response: We thank the reviewer for pointing out that even without XRD data, we employed a variety of advanced spectroscopic techniques, as well as calculation to support the structure of the trefoil knot.

Comment: The evidence here is very compelling, but it is also conceivable (but unlikely) that the product could be a 2+1 [2]catenane in fast exchange (like the Hunter/Vögtle knot story - *Angew. Chem. Int. Ed.*, 2000, 39, 1616). Did the authors check the MS/MS fragmentation pattern to confirm this is really a knot rather than a catenane? If not, that experiment could be good to perform. It's very unlikely that the structure is anything except a trefoil knot, but it is a simple experiment to rule out another explanation.

Response: We agree with the reviewer that even though evidence shows that the trefoil knot is the most possible product, other possibility such as 2+1 catenane should also be ruled out, in the case that single crystal structure is not obtained.

First of all, the possibility of the 2+2 homo-catenane (that is a [4+8] product) is ruled out in the last question discussed above. The 2+1 catenane that the reviewer proposed here is a catenane composed of a bigger macrocyclic component that contains two tetraaldehyde and four bisamine precursors, which is encircled by a smaller ring containing one tetraaldehyde and two bisamine precursors. This catenane and the trefoil knot would have identical molecular ion peaks in the mass spectrum. In fact, this possibility can easily be ruled out by using ¹H NMR spectroscopic results.

First of all, the two macrocyclic components with different sizes should have different chemical environments. As a consequence, in the ¹H NMR spectrum, this [2]catenane should exhibit two different set of resonances whose integrations should be 2:1. However, the ¹H NMR spectrum of our product only shows one set of resonances, indicating that this molecule has a highly symmetric architecture.

Second, it is noteworthy that the formation of the smaller macrocycle, namely the one containing one tetraaldehyde and two bisamine precursors, is not even geometrically possible. The distance between two methylene units in the biphenyl unit is significantly longer than the two amino units in a CHDA linker.

Comment: The CD data of the knot enantiomers is nice, but a missing piece of data that would be very good to have for comparison is the CD of the enantiomers of the macrocycle **3**. Have the authors measured this?

Response: We did record the CD of the two enantiomers of macrocycle **3**, which were added both in the SI and here. The CD spectra of **3** (0.06 mM) are different from those of trefoil knots **2**.

Comment: Have the authors tried cooling the solution to freeze out the rotation of protons g and f? It would be very interesting to see the energetic barrier to this rotation, as it could tell a lot about the knot tightness.

Response: We have recorded the ^1H NMR spectra at lower temperature namely -20°C . 25% Acetone- d_6 was added into the aqueous solution, in order to avoid freezing. However, the number of the resonances of the trefoil knot remained nearly unchanged, even though all of the peaks became a little broader. Such observation indicates that the rotation of biphenyl unit in the trefoil knot is still relatively fast at -20°C . Therefore, we cannot use VT NMR spectroscopy to measure the energy barrier of this rotation. Further lower the temperature led to freezing of the aqueous solution.

Comment: The references need some additional work. Nitschke's recent imine knot (Chem, 2021, 1534) should definitely be cited, as should some of the pioneering knot work from Vögtle (rather than the example from Feigel which is very interesting but was derived from Vögtle's lab). Also, an intro sentence say "In some cases, these topologically complicated architectures are related to their biological activity^{5,8-10}" but in ref 8 the topology has nothing to do with the biological activity and ref 9 and 10 don't concern biology at all (they're highly relevant papers to cite though, just not in this context).

Response: We thank the reviewer for the suggestions. We added these papers in the reference part as requested. See the newly-added references 25, 29, 30.

And we also modified ref 8,9 and 10, as requested. Only reference 5 is cited as an example of biological related knots.

Comment: The term “Solomon knot” should be changed to “Solomon link” .

Response: We thank the reviewer for reading the paper so carefully and pointing out our mistake. We have modified the main text accordingly.

Comment: Sentence “By using the Stokes – Einstein equation, the hydrodynamic diameter of S-26+ was calculated to be approximately 2.6 nm” – maybe compare this value with the diameter calculated from the molecular model?

Response: The radius obtained from the molecular model is defined by the distance between the center point of the knot and one of the marginal atoms, which is 13.2 Å. Therefore, the diameter calculated from the molecular model is about 26.4 Å (13.2 Å X 2), which is consistent with our experimental result 2.6 nm. And we have added this comparison in the SI.

Comment: Sentence “In the framework of S-26+, the protons e/e', c/c' and h/h' become diastereotopic in the 1H NMR spectrum, revealing the chirality of the knot.” As per the authors own discussion later, isn' t the h/h' splitting due to endo/exo effects rather than diastereotopicity?

Response: We thank the reviewer for pointing out our mistake. We have modified the main text accordingly.

Comment: Sentence “This observation indicates unambiguously that S-34+ is a topologically trivial molecule.” I don' t agree – NOESY is notoriously sensitive to dynamics so a topologically complex molecule in fast exchange could still lack NOESY signals. Rephrase?

Response: We thank the reviewer for pointing out our misrepresentation. We agree with the reviewer that lacking of NOESY signal cannot unambiguously prove the trivial architecture of the macrocycle. In fact, this experiment showing that S-2⁶⁺ and S-3⁴⁺ have different NOESY spectra, can prove that the macrocycle and the trefoil knot have totally different structures. We have modified the main text accordingly. In the modified version of the maintext, our description is as follows, “This observation is consistent with our hypothesis that, unlike the structure of S-2⁶⁺, S-3⁴⁺ does not have an intertwined architecture.”

Comment: SI: DMF is not a very good internal standard due to its low boiling point (relatively speaking). Over longer time frames and at elevated temperatures you can lose a bit of DMF due to evaporation, hence giving you inaccurate readings.

Response: We agree with reviewer that DMF might undergo evaporation so that the yield calculation

might become inaccurate.

We thus performed a control experiment which confirms that DMF would not undergo remarkable evaporation to a significant extent in our experimental condition. That is, we combined the same amount of DMSO and DMF in D₂O placed in a sealed NMR tube. This solution was heated for 12 h at 80 ° C, which mimics our self-assembly condition. We recorded the ¹H NMR spectra before and after heating the solutions. The spectra indicated that relative integration of DMF/DMSO remained unchanged. Assuming that DMSO is a solvent which can hardly escape from the tube, due to its high boiling point, it is acceptable to assume that DMF did not undergo significant evaporation during the self-assembly. Therefore, DMF can be used as the internal standard.

Reviewer 2:

Comment: The manuscript submitted by Hao Li et al. reports the hydrophobically-driven synthesis of a trefoil knot. Overall, the study is well done and the conclusions appropriately drawn. My main issue with this manuscript concerns the novelty of the work. In my opinion, the most important findings of the study have already been reported elsewhere. If I go through the abstract: (1) “Herein, we report a rare case of a trefoil knot that can be self-assembled efficiently in water without involving additional templates.” The hydrophobically-driven synthesis of trefoil knots and other topologically complex molecules have been reported in ref. 31, 32, 33 and 34.

Response: We totally agree with the reviewer that our results are performed by obtaining the inspiration from works of other scientists, as cited in ref. 31, 32, 33 and 34. And surely it is not the first time that hydrophobic effect is used as the driving force of the formation of knots without using metal templates. However, our work, for the first time, employed typical imine bond in self-assembling trefoil knot in water. This success encourages chemists to use imine condensation as a general method to self-assemble all kinds of molecules with complex architectures, not just cages and catenanes that were reported before. The second merit is that, in our trefoil knot systems, the handedness is precisely controlled by a commercially available chiral bisamine, namely R or S- CHDA, instead of a sophisticatedly synthesized chiral compound that used in Sanders's trefoil knot.

We changed the phrase “a rare case of a trefoil knot” to “a trefoil”, implying that our knot is not the first example that could self-assembly without external templates.

Comment: (2) “The direct condensation of three equivalents of a tetraformyl precursor and six equivalents of a chiral diamine produced successfully a [3+6] trefoil knot whose intrinsic handedness is dictated by the stereochemical configuration of the diamine linkers.” This effect is well-established. It has been known since the pioneering work of Sauvage (ACIE 2004, 43, 4482) and has been observed numerous time. Particularly relevant examples include Chem 2021, 7, 6, 1541 and ref. 13-14.

Response: We agree with the reviewer that it is a well-established method to use a chiral building block to control the handedness of a self-assembled molecules. We thank the reviewer for bringing a few papers including the one by Sauvage to our attention, which was added in the reference part. In the modified version of the main-text, we added the following sentence “It is noteworthy that the approach of using a building block with stereo chiral centers to dictate the handedness of a topological complex molecule were also reported by many other groups^{25,34}, including the one by Sauvage⁵⁰.” In fact, we never argue that this is a milestone-level discovery. We just want to point out that one of our merits is that we are able to use a commercially available chiral building block, to precisely control the handedness of the self-assembled trefoil knot.

Comment: (3) “When a racemic mixture of the bisamine linker was used in the reaction, narcissistic self-sorting occurred, yielding a pair of enantiomeric trefoil knots, each of which contains six linker units with the same stereochemical configuration.” The most relevant, previously published, example of this behaviour is perhaps ref. 34.

Response: We agree with the review that Sander's work (ref 34) also successfully realized narcissistic self-assembly. Again, we totally concede that our work got inspiration from the works of Sanders and others. In fact, we never argued such results represent a milestone-level discovery. In the modified version of the maintext, we added a sentence that “Such narcissistic self-sorting behavior was also observed by other scientists such as Sanders³⁴.”

Comment: (4) “Contrary to the conventional wisdom that water is not compatible with the dynamic covalent chemistry of imine condensation, the multivalent cooperativity between all the imine bonds within the framework makes this trefoil knot robust in the aqueous environment.” This argument was already the selling point of another article published last year by the same group (ref. 46). The idea can be traced back to the work of Warmuth on water soluble imine-based cryptophanes (ref 47-48). Along the same lines, the conclusion reads: “This work also overturns the commonly accepted precept that imine-based dynamic covalent chemistry cannot be performed in the presence of water, which would encourage chemists to self-assemble other topologically complex molecule in aqueous media.” However, imine bonds were already used for the synthesis of interlocked molecules in pure water in Chem. Sci. 2018, 9, 1317, and the argument of this previous study was identical. This study also provides an analysis of the effect of gauche interactions on the assembly of interlocked molecules that seems relevant to the current findings (see “... indicated that restricted gauche conformations...” p. 6).

Response: We thank the reviewer for reading the paper so carefully. We also thank the reviewer for pointing out that we did not cite an important paper by Cougnon, namely Chem. Sci. 2018, 9, 1317. This paper has been added as ref 49. And we have changed the description in main-text, “This work further supports that imine-based dynamic covalent chemistry can be performed in the presence of water, which would encourage chemists to self-assemble other topologically complex molecule in aqueous media.” In addition, as mentioned by the reviewer, the importance of gauche effect was also discussed in Chem. Sci. We added a sentence to acknowledge this discovery in the modified version, namely “The importance of *gauche* effect in determining the distribution of self-assembly products was also reported by other scientists such as Cougnon⁴⁹”.

It is noteworthy that even though using water as the solvent to perform imine condensation is reported, it is still a well-accepted precept that water should be avoided in self-assembly using imine. In addition, even though water-based imine self-assembly is not fully novel, synthesizing an imine-based trefoil knot whose chirality is precisely controlled, is still rare or not reported. This discovery encourages chemists to use imine in water as a general dynamic bond to self-assemble all kinds of molecules, not only cages and catenanes, but also knots.

Comment: (5) “Furthermore, the presence of water was proven to be essential for the trefoil knot formation.” See most of the references mentioned above.

Response: Again, we concede this result is not milestone-level discovery.

Comment: What exactly do we learn from this paper that was not previously published? This manuscript present an extremely interesting finding, but I believe the authors did not make the most of it and have confined their work to familiar ground. Therefore I would not recommend publication of this work, as it is now, at the level of Nature Communications.

Response: We thank the reviewer for reading the paper so carefully. We totally understand that our results are not milestone-level discovery, including i) imine-based self-assembly in water is not the first report; ii) using hydrophobic effect to drive knot formation is not new either; iii) using chiral building block to precisely control the handedness of a self-assembled molecule is not new.

However, our trefoil knot is the first example that using imine condensation to self-assemble a trefoil knot

whose chirality can be precisely controlled by commercially available chiral starting materials.

Comment: This study has also raised a few questions: (1) What is the effect of temperature on the process of self-assembly and on the hydrolysis of the imine bond (which is usually rapid at high temperature)? The data presented is rather scarce (Figure S22-24) and is not quantitative. It is difficult to draw any conclusion relative to the kinetics of formation of S-2 or to the kinetics of formation/hydrolysis of the imine bond.

Response: Temperature did affect both the kinetics and thermodynamics of the trefoil knot formation.

First, in terms of kinetics, self-assembly from the aldehyde/amine precursors to the trefoil knot product S-2⁶⁺ or R-2⁶⁺ became faster/slower at higher/lower temperature. For example, at room temperature, the conversion took a few days to completion. At 80 °C, the self-assembly system reached its equilibrium within a few hours.

Considering 1²⁺·2Br⁻ is not fully soluble in water at 80 °C, we performed counterion exchange and obtained a more water-soluble precursor namely 1²⁺·2CH₃COO⁻. After the amine precursor namely (SS)-CHDA was added into the water solution of 1²⁺·2CH₃COO⁻, we heated the solution at 80 °C. The ¹H NMR spectra of the sample were recorded at different amount of time. One of the resonances (marked with a red arrow) corresponding to a proton in the trefoil knot was integrated (Figure S25 or below), so that we are able to obtain a plot of the yield of the trefoil knot product versus time (Figure S26 or below). It was clearly shown that the trefoil knot formation was complete within 100 min, when the solution was heated at 80 °C. However, we discovered that S-2⁶⁺·2CH₃COO⁻ would undergo decomposition partially even at room temperature due to the basicity of CH₃COO⁻, which means that the yields would be more than what we calculated.

Second, in terms of thermodynamics, higher temperature disfavors the imine formation reaction, shifting the equilibrium to the side of aldehyde and amine precursors. This is not surprising, given that formation of imine is exothermic. VT ^1H NMR spectroscopic experiments support this argument (see the figure attached below). For example, when the temperature goes up to 90°C , in the ^1H NMR spectrum of aqueous solution of S-2^{6+} , the resonances corresponding to the aldehyde precursor was also observed. After cooling the solution to room temperature, the resonances corresponding to aldehyde precursor disappeared, and the trefoil knot became the only observable compound, indicating that at lower temperature, the equilibrium shifted to the side of imine formation.

Comment: (2) The variable temperature of the trefoil knot was only run up to 60°C while that of the macrocycle was ran up to 110°C (Figure 23-24). Any reason for that?

Response:

First, the macrocycle was self-assembled in DMSO, an organic solvent where imine bond is relatively more robust. Heating the solution of the Hmacrocycle up to 100°C did not lead to macrocycle

decomposition. In the case of the trefoil knot whose formation occurred in water, the imine bonds are more labile. Raising temperature to 80 °C would lead to imine hydrolysis partially.

Second, the major purpose of VT experiment is to justify the rate of the intramolecular flipping motion. In the case of the macrocycle, it underwent intramolecular flipping. We recorded its ^1H NMR spectra from room temperature to 110 °C, in order to accurately determine the flipping kinetics. As a comparison, the trefoil knot has an intertwined structure, and therefore it did not undergo such intramolecular motion. The ^1H NMR spectrum of the trefoil knot did not undergo remarkable changes when we raised the temperature. In fact, we did record its ^1H NMR spectrum at 90 °C. Its ^1H NMR spectrum is almost the same as that recorded at room temperature, except that imine hydrolysis was observed to some extent. See the figure above.

Comment: (3) What is the effect of concentration of the assembly? The trefoil knot is more likely to hydrolyse at low concentration. Yet, the authors managed to measure a CD spectrum. I could not find the concentration at which this spectrum has been recorded, but I can imagine it's two to three orders of magnitude lower than that of the NMR experiments. Is the trefoil knot also stable in dilute conditions over time?

Response: The reviewer was totally correct in the statement that lower concentration might lead to trefoil knot decomposition. This is indeed the case, namely that we observed that when the concentration is relatively low (e.g., 0.5 mM), the self-assembly of the trefoil knot became unsuccessful. However, the trefoil knot is rather kinetically inert for a substantial amount of time. For example, the trefoil knot was successfully obtained at a relatively higher concentration namely 1 mM. Then we diluted the solution to 0.04 mM and recorded the ^1H NMR spectrum. We observed that the ^1H NMR spectrum of this trefoil knot did not change within hours. It will decompose gradually, after two or three days. It is noteworthy that even after three days, the trefoil knot is not fully hydrolyzed, indicating that the knot is remarkably kinetically inert. Considering the CD experiments were performed with 20 min after dilution, we reasonably propose that it would not decompose during this CD experiment, and therefore the CD spectrum should be assigned to the trefoil knot, instead of the decomposition product.

Comment: (4) Knowing the expertise of the Li group in anion recognition, it is somewhat surprising to see that the role of the counter ion in the assembly of S-2 has not been investigated. Based on the DFT calculations, it looks like the trefoil knot could possess a central cavity with several converging C-H hydrogen bond donors. I am not sure whether this cavity is large enough to accommodate bromide. Nevertheless, the observation that the trefoil knot flies with one and two bromides in MS (Figure S13), while the trivial macrocycle does not (Figure 21), suggests that two equivalents of bromides could be bound in the cavity (see Chem. Sci. 2016, 7, 2524 for a previous example). Does bromide (or other anions) bind to the knot? Does it template its formation?

Response: We thank the reviewer for this suggestion.

First, after trefoil knot formation, we added a few types of salt, including $\text{TBA}^+\cdot\text{Cl}^-$, $\text{TBA}^+\cdot\text{Br}^-$, $\text{TBA}^+\cdot\text{I}^-$ and $\text{TBA}^+\cdot\text{NO}_3^-$ to a $\text{D}_2\text{O}/\text{DMSO-}d_6$ solutions of S-2^{6+} . We found that addition of these anions did not lead to any changes in the ^1H NMR spectrum of the trefoil knot, whose counteranion is Br^- .

Second, we employed tetraaldehyde precursor with other counter-anion for self-assembling the trefoil knot, namely CH_3COO^- . The trefoil knot $\text{S-2}^{6+}\cdot 6\text{CH}_3\text{COO}^-$ was successfully obtained, in the absence of Br^- anion. The corresponding ^1H NMR spectrum of $\text{S-2}^{6+}\cdot 6\text{CH}_3\text{COO}^-$ is almost identical as that of $\text{S-2}^{6+}\cdot 6\text{Br}^-$. We also added $\text{TBA}^+\cdot\text{Br}^-$ into the aqueous solution $\text{S-2}^{6+}\cdot 6\text{CH}_3\text{COO}^-$, whose ^1H NMR spectrum barely changed (See figure below).

Such observations clearly indicated that the trefoil knot cannot recognize any anions including Br^- in water. And therefore, Br^- did not template the formation of the trefoil knot.

Comment: (5) Imine bond hydrolysis strongly depends on pH. In the manuscript, here is no mention of the pH of the solution during the reaction. Does pH affect the assembly?

Response: We performed the self-assembly of the trefoil in different pH conditions. The results indicated that trefoil knot only forms in neutral to weakly basic conditions. In acidic condition, imine bonds began to decompose. In the condition of $pD = 4$, the trefoil knot partially hydrolyzes into its aldehyde precursor. Further acidifying the solution led to complete hydrolysis. In strong basic condition ($pD > 10$), the compounds underwent decomposition partially, because pyridinium units and aldehyde functional groups are not stable in such condition. For example, aldehyde undergoes Cannizzaro reaction in strong basic aqueous media.

Comment: (6) While I am convinced by the knotted topology of S-2I wonder whether it is possible to be sure that S-2²⁺ is not a trivial macrocycle folded in a chiral conformation.

Response: The reviewer is correct in the statement that a trivial macrocycle folded in a chiral conformation is also likely to occur. The truth is that, however, this potential “foldamer” should exhibit a ¹H NMR spectrum that is temperature dependent. In our case, the trefoil knot provided a temperature independent ¹H NMR spectrum from 25 °C to 90 °C, even though at higher temperature the knot underwent partial hydrolysis. These temperature-independent spectra are obviously inconsistent with the proposed folded macrocycle. As a comparison, in the case of macrocycle S-3⁴⁺, the ¹H NMR spectrum was highly temperature dependent.

In addition, for a potential foldamer without an intertwined structure, the intramolecular CH- π interactions are relatively weak and its conformation is less preorganized. The resonances corresponding to the protons g and f in the putative macrocycle would not undergo such remarkable upfield shifts, such as 2 ppm.

Comment: There are also a few mistakes that must be corrected before publication. (1) p. 4: “the protons e/e', c/c' and h/h' become diastereotopic in the ¹H NMR spectrum, revealing the chirality of the knot”. Protons e/e' are diastereotopic. Protons c/c' and h/h' are not diastereotopic, they are simply inequivalent because the two imine linkages adopt different conformations (exo and endo). This inequivalence has nothing to do with the chirality of the trefoil knot.

Response: We thank the reviewer for reading the paper so carefully and pointing out our mistake. We have modified the main text accordingly

Comment: (2) p. 4 “Only one resonance was observed corresponding to proton g and f, indicating that the biphenyl units in the framework of $S-2^{6+}$ undergo fast circumvolution around the –biphenyl– axis on the ^1H NMR timescale.” Whether it occurs rapidly or not, circumrotation around the biphenyl axis does not exchange the protons g and f and cannot explain this (incidental?) equivalence.

Response: We think we might make a misunderstanding here. See the following figure. Within the framework of the trefoil knot, f1/f2, as well as g1/g2 should have different chemical environment, respectively. Fast circumvolution around the –biphenyl– axis makes f1=f2, and g1=g2 in the ^1H NMR spectrum. In the modified version of the maintext, we changed the description, “Only one resonance was observed corresponding to the proton including either g or f respectively, indicating that the biphenyl units in the framework of $S-2^{6+}$ undergo fast circumvolution around the –biphenyl– axis on the ^1H NMR timescale.”

Reviewer 3:

Comment: From the perspective of theoretical calculation, this is a nice work to assist the experiment to explain the mechanism behind the experimental phenomenon. However, we know that for such a system, since the rotation of the dihedral angle can produce many possible conformations, how can the author ensure that the present structure is representative of the real experimental system? Details are needed here, including how many possible structures are built and how to eliminate some of the impossible ones.

Response:

We totally agree with the reviewer that the rotation of the dihedral angles can produce many possible conformations. The truth is that, even although in the framework of the trefoil, many possible conformations might be possible due to rotation of the C-C single bond and the dihedral angle, many of unreasonable or impossible conformations could actually be ruled out by our ^1H NMR spectroscopic data. For example, the highly symmetric nature of the ^1H NMR spectrum indicates that the trefoil knot has a C_3 symmetric structure. NOE signals between the biphenyl protons and the protons in the phthalaldehyde residues supported the intertwined architecture. NOESY spectrum indicated that in each phthalaldehyde residue, the two imine units have two different orientations namely *endo* and *exo*. As a consequence, many impossible conformations can be ruled out here. For example, the trefoil knot with *endo-endo*, or *exo-exo* imine conformations could be unambiguously eliminated. ^1H NMR spectrum can also provide information to indicate that, many units including the biphenyl and pyridinium undergo fast circumvolution. For example, the protons *g*, *f* in the biphenyl, as well as the pyridinium protons *a* and *b*, all have only one relatively sharp resonance in the ^1H NMR spectrum.

Based on these experimental results, the trefoil $\mathbf{S-2}^{6+}$ is supposed to have four possible diastereoisomers, including $\mathbf{P-\bar{A}-S-2}^{6+}$, $\mathbf{M-\bar{A}-S-2}^{6+}$, $\mathbf{M-A-S-2}^{6+}$ and $\mathbf{P-A-S-2}^{6+}$, whose enantiomers are $\mathbf{M-A-R-2}^{6+}$, $\mathbf{P-A-R-2}^{6+}$, $\mathbf{P-\bar{A}-R-2}^{6+}$ and $\mathbf{M-\bar{A}-R-2}^{6+}$, respectively. In each of these possible diastereoisomers, each phthalaldehyde residue has the same imine orientation manner, namely either *endo-exo*, or *exo-endo*.

We then employed DFT calculation to predict the corresponding relative free energies of all four possible diastereoisomers, namely $\mathbf{P-\bar{A}-S-2}^{6+}$, $\mathbf{M-\bar{A}-S-2}^{6+}$, $\mathbf{M-A-S-2}^{6+}$ and $\mathbf{P-A-S-2}^{6+}$. $\mathbf{P-\bar{A}-S-2}^{6+}$ among these four structures is found to represent the most thermodynamically favored one. Meanwhile, as shown in the following figures a) and b), DFT calculations are able to predict the theoretical UV-Vis and CD spectra of $\mathbf{P-\bar{A}-S-2}^{6+}$, which are consistent with experimental ones. We also use DFT to predict the CD spectra (see figure d) of $\mathbf{M-\bar{A}-S-2}^{6+}$, $\mathbf{M-A-S-2}^{6+}$ and $\mathbf{P-A-S-2}^{6+}$. However, these predicted spectra are very different from the experimental one. Thereby, it is reasonably to believe that $\mathbf{P-\bar{A}-S-2}^{6+}$ should be the real product. In the revised Supporting Information, we have updated figure S30 to show the theoretically predicted UV-Vis and CD spectra for all the four diastereoisomers.

Comment: In addition, the relative free energy is more meaningful than the relative energy, and the calculation of frequency can verify whether the structure is the local minimum point of the potential energy surface.

Response: We thank the reviewer for the helpful suggestions. We have performed additional frequency and free energy calculations. As shown in the table below, although $P\text{-}\bar{A}\text{-}S\text{-}2^{6+}$ and $P\text{-}\bar{A}\text{-}R\text{-}2^{6+}$ have one imaginary frequency, the imaginary frequency is only -8 cm^{-1} , which is already a very small number for such a large molecule containing hundreds of atoms. Thereby, it is reasonable to use these four optimized geometries to predict UV-Vis and CD spectra and compare with the experimental observations. Besides, the relative stabilities of the four investigated diastereoisomers based on the relative free energies are the same as those based on relative energies. Namely, the most stable diastereoisomer is still $P\text{-}\bar{A}\text{-}S\text{-}2^{6+}$. Note that if the imaginary frequency can be eliminated, its energy could be further reduced and becomes even more stable than the other three diastereoisomers. In the revised manuscript, we have replaced the relative energies to relative free energies.

	ΔG (kcal/mol)	number of imaginary frequencies	imaginary frequencies (cm^{-1})
$P\text{-}\bar{A}\text{-}S\text{-}2^{6+}$	0	1	-8
$P\text{-}A\text{-}R\text{-}2^{6+}$	17.6	0	0
$P\text{-}\bar{A}\text{-}R\text{-}2^{6+}$	25.6	1	-8
$P\text{-}A\text{-}S\text{-}2^{6+}$	39.4	0	0

REVIEWERS' COMMENTS

Reviewer #1 (Remarks to the Author):

I would like to thank the authors for carefully considering all my comments and suggestions, and for doing their best in addressing the concerns I had with the previous version of the manuscript. From a technical perspective, this is an excellent update as the authors have convincingly addressed all my scientific questions. The updates to the SI are good strong and I think the authors have done a decent job with the main MS as well. However, I still believe the reporting is too "loose" and unspecific in terms of the (lack of) isolation of the knot. The authors should state clearly in the main MS that the knot is generated in situ and cannot be isolated, nor reduced with NaBH₄ or otherwise derivatized – this is very important information for the community to know when drawing upon these design rules to design new systems.

That it is not possible to isolate or derivatize the knot in any way is definitely a significant problem, as it will greatly limit the usefulness of the system. As such, the system loses a bit of its appeal. I would be delighted if the authors could make this work with oximes/hydrazones (though it would then very closely resemble the Cougnon system), but given how specific the building blocks and conditions seems to be I suspect this will not be simple. I partially agree with referee 2 that there is much here that has been directly inspired from other work, but given that the authors have addressed all my concerns and I still think the paper has many important insights in terms of building block design for knotting, I will still recommend this for publication.

Reviewer #2 (Remarks to the Author):

I have read the other referees' comments and the authors' response with great interest. The authors have appropriately addressed all the comments and I can only repeat that it is a very neat study, further strengthened by the new experiments added to the SI. There are only a few reports of trefoil knots synthesized using the hydrophobic effect, which makes this work attractive. I do remain skeptical about the novelty of this study relative to the previous ones, but I agree that the use of imine chemistry in water is interesting and presents some advantages (it is notably possible to use a commercially available chiral diamine). As a side note, the hydrophobically-driven formation of imine bonds is not as uncommon as the authors claim. Imine bonds have not only been used to assemble cages and catenanes in water, but also micelles (JACS 2009, 131, 11274; Macromolecules 2011, 44, 1327; Chem. Eur. J. 2021, 27, 13457), dynamers (JACS 2012, 134, 4177) and other supramolecular polymers (Science 2016, 351, 497; Nat. Commun. 2019, 10:450) – to cite only a few examples.

Reviewer #3 (Remarks to the Author):

I am very satisfied with the author's answers and revisions, and I suggest that this manuscript is can be accepted for publication

Letter of Response to the Reviewers' Comments

Reviewer #1 (Remarks to the Author):

I would like to thank the authors for carefully considering all my comments and suggestions, and for doing their best in addressing the concerns I had with the previous version of the manuscript. From a technical perspective, this is an excellent update as the authors have convincingly addressed all my scientific questions. The updates to the SI are good strong and I think the authors have done a decent job with the main MS as well. However, I still believe the reporting is too "loose" and unspecific in terms of the (lack of) isolation of the knot. The authors should state clearly in the main MS that the knot is generated in situ and cannot be isolated, nor reduced with NaBH₄ or otherwise derivatized – this is very important information for the community to know when drawing upon these design rules to design new systems.

That it is not possible to isolate or derivatize the knot in any way is definitely a significant problem, as it will greatly limit the usefulness of the system. As such, the system loses a bit of its appeal. I would be delighted if the authors could make this work with oximes/hydrazones (though it would then very closely resemble the Cougnon system), but given how specific the building blocks and conditions seems to be I suspect this will not be simple. I partially agree with referee 2 that there is much here that has been directly inspired from other work, but given that the authors have addressed all my concerns and I still think the paper has many important insights in terms of building block design for knotting, I will still recommend this for publication.

Response: We have revised the main text and discussed the labile nature of this trefoil knot. The following sentences were added, "The attempts to isolate the trefoil knot **S-2**⁶⁺ via counterion exchange were unsuccessful, probably because imine bonds underwent degradation during these processes. We also tried to fix the architecture of the trefoil knot by means of imine reduction. Such trials were not successful either, because under such condition, the cationic pyridinium building blocks became labile."

Reviewer #2 (Remarks to the Author):

I have read the other referees' comments and the authors' response with great interest. The authors have appropriately addressed all the comments and I can only repeat that it is a very neat study, further strengthened by the new experiments added to the SI. There are only a few reports of trefoil knots synthesized using the hydrophobic effect, which makes this work attractive. I do remain skeptical about the novelty of this study relative to the previous ones, but I agree that the use of imine chemistry in water is interesting and

presents some advantages (it is notably possible to use a commercially available chiral diamine). As a side note, the hydrophobically-driven formation of imine bonds is not as uncommon as the authors claim. Imine bonds have not only been used to assemble cages and catenanes in water, but also micelles (JACS 2009, 131, 11274; Macromolecules 2011, 44, 1327; Chem. Eur. J. 2021, 27, 13457), dynamers (JACS 2012, 134, 4177) and other supramolecular polymers (Science 2016, 351, 497; Nat. Commun. 2019, 10:450) – to cite only a few examples.

Response:

We thank the reviewer again for agreeing publishing this paper.

We also thank the reviewer for bringing so many examples of using imine to do synthesis in water to our attention. It is noteworthy that, even though imine reaction is indeed used a lot in synthesizing micelles and polymers as mentioned by the reviewer, using imine as the reaction motif to self-assemble small molecules with well-defined structures is still rare. We modified abstract. Here, we claimed that “imine is not amenable to use in water”, instead of the previous argument that “water is not compatible with the dynamic covalent chemistry of imine condensation”. We also cited the paper mentioned by reviewer 2, indicating that imine condensation is widely used in micelle, dynamers, and supramolecular polymer.

Reviewer #3 (Remarks to the Author):

I am very satisfied with the author's answers and revisions, and I suggest that this manuscript is can be accepted for publication

Response: We thank the reviewer for providing such a relatively positive comment.